# The Dual Nature of Amaranth—Functional Food and Potential Medicine

**DOI:** 10.3390/foods11040618

**Published:** 2022-02-21

**Authors:** Justyna Baraniak, Małgorzata Kania-Dobrowolska

**Affiliations:** Department of Pharmacology and Phytochemistry, Institute of Natural Fibres and Medicinal Plants National Research Institute, Wojska Polskiego 71b Str., 60-630 Poznan, Poland; malgorzata.kania@iwnirz.pl

**Keywords:** amaranth, pseudo-cereals, functional food, biological activity, pharmacological activity, health benefits

## Abstract

The beneficial health-promoting properties of plants have been known to mankind for generations. Preparations from them are used to create recipes for dietary supplements, functional food, and medicinal products. Recently, amaranth has become an area of increasing scientific and industrial interest. This is due to its valuable biological properties, rich phytochemical composition, and wide pharmacological activity. Amaranth is a pseudo-cereal crop with a dual character, combining the features of food and health-promoting product. This paper briefly and concisely reviews the current information on the chemical composition of amaranth, the value of its supplementation, the status of amaranth as a food ingredient as well as its key biological and pharmacological activities. The beneficial biological properties of amaranth preparations described in this paper may be an incentive to conduct further in-depth scientific research in this field and also to promote the development of innovative technologies in the food and cosmetics industry with the use of this plant.

## 1. Introduction

In recent years, there has been noticed a growing interest in plant raw materials whose properties allow them to be used in both food and medicines. Various cereal grains are widely used in the food and beverage industry. There is a fairly broad group of plants that are classified as so-called pseudo-cereals. This means that the edible parts of these plants are the seeds and they usually are consumed in a similar way to cereals, being processed into flour. They also have similar nutritional values and taste to cereals. These are not typical cereals, but due to their similar composition and nutritional value mentioned above, they can be a good alternative. Pseudo-cereals have been the staple food of our ancestors for thousands of years, and all over the world. In different regions of the world, different pseudocereals predominate. Even today, pseudo-cereals still form the basis of nutrition in the poorest parts of the world. They have been increasingly appreciated in European countries for a long time. The best-known pseudo-cereals are amaranth, buckwheat, sorghum, millet, chia as well as khorasan. Actually, the most widely studied pseudocereals are quinoa, amaranth, chia, and buckwheat [1]. They present great potential as a natural source of a wide spectrum of biologically active compounds. Recent work suggests that first and foremost peptides and protein hydrolysates derived from these beneficial species for the human health are worth considering [1]. The first study on an amaranth protein deriving bioactive peptide with cholesterol esterase and pancreatic lipase inhibitory activities was published in 2021 by Ajayi and colleagues [2].

Due to climate change, the problem of world hunger, and changes in crop profiles in European and other countries around the world, it is becoming desirable to look for new plants with a high nutritional potential that can be combined with health benefits. Amaranth is a plant with valuable qualities as food and additionally with many valuable health properties. Additional important advantages of this plant are satisfactory yield performance, drought resistance, and enhanced photosynthesis. The excellent nutritional value of amaranth [3], the diverse chemical composition of amaranth seeds and leaves, the wide spectrum of biological activity, health-promoting properties, and the pharmacological activity of the plant have aroused the interest of researchers in recent years. This has resulted in a significant increase in the number of scientific studies on the properties and potential use of preparations from this plant. The PubMed database was used to locate publications with the most important data describing the nutritional and pharmacological activity of amaranth preparations.

## 2. Plant Characteristics

Amaranth has been well known since the time of the Aztecs, Mayans, and Incas [4]. In the 16–17th centuries, it spread widely in various other countries as a cereal, vegetable, weed, or crop. Amaranth seeds were used as food, but also as a sacred plant. It was used in many religious and ritual ceremonies [5]. It is a valuable plant whose potential is still not sufficiently exploited. This should be clearly emphasized because it has a huge economic value due to the various benefits it can bring to producers, food processors, and consumers. Amaranth is a member of the *Amaranthaceae* family comprising about 70 species of annual plants [4,6,7,8,9]. In many countries, *Amaranthus* species are cultivated for use as cereals, vegetables, or ornamentals, a few species are considered weeds. A review of the current literature suggests that mainly *Amaranthus cruentus, Amaranthus hypochondriacus*, and *Amaranthus caudatus* are grown for food purposes [4,9,10]. *Amaranthus blitum* Linn., *Amaranthus gangeticus* Linn., *Amaranthus mangostanus* Linn., *Amaranthus tricolor* Linn. Are cultivated all over India as a vegetable. Amaranth leaves are used in salads and to prepare other dishes, in African countries amaranth leaves are sometimes recommended for medicinal purposes [9]. Other species of amaranth, such as *A. viridis, A. tricolor, A. retroflexus,* and *A. hybridus* are known mainly as a vegetable. These species of *Amaranthus* grow very well in hot and humid regions of our globe. In Poland, amaranth is cultivated for seeds as a source of lipids and proteins for the production of flour, flakes, confectionery, expanded grains and bread, pasta, and noodles [10]. *Amaranthus cruentus* is the most widely grown species of this plant genus [5].

## 3. Chemical Composition of Amaranth

The main biological compounds found in amaranth are proteins, fats, carbohydrates, vitamins, and minerals [8]. The protein content (~18%) of amaranth seeds is higher than that of traditional cereals and varies according to the variety of the plant, the climate, and soil conditions and the method of fertilization [7,10]. Among proteins, albumins are the largest fraction. Protein contains all the essential amino acids required by the body [6], especially a lot of lysine and tryptophan. Starch is the main carbohydrate found in amaranth [7]. The amount of starch in amaranth seeds is approximately 45–65% [10]. An important group of compounds found in amaranth is the fiber fraction (high level)—its soluble (mainly pectins) and insoluble parts. The insoluble fraction consists of lignin, cellulose, and hemicelluloses, which have a beneficial effect on the digestive system. The amount of fiber in seeds, depending on the source of origin, averages 2–8% of dry weight [5]. The nutritional value of amaranth seed is mainly caused by lipids (~7%) [5] with a good ratio between saturated and unsaturated fatty acids and high protein content with the essential amino acids composition better than that in FAO/WHO standards [3,10]. Among unsaturated fatty acids, the most abundant are linoleic (~62%), oleic (~20%), linolenic (~1%), and arachidonic acid [5,11]. Amaranth contains saturated fatty acids (palmitic (~13%), stearic (~2.6%), arachidic (~0.7%), and myristic (~0.1%) in small amounts [5]. Among the lipid fraction of amaranth, tocopherols, tocotrienols, and sterols play an important biological role [12]. Squalene has been identified in the seeds and leaves of the plant, and they are also very rich in vitamins (especially the B group) and minerals [8]. The percentage content of squalene in oil derived from amaranth is 2–8% [6] or 6–8% [5,13], depending on the source and author. Amaranth seeds are a very good source of minerals, representing an average of 3.3% of their weight [10]. The levels of calcium, potassium, and magnesium are quite high, with iron, phosphorus present in the largest amount. Other minerals identified in amaranth include copper, zinc, sodium, chromium, manganese, nickel, lead, cadmium, and cobalt. The seeds and leaves of amaranth contain small amounts of polyphenols, saponins, hemagglutinins, phytin and nitrates (V), and oxalates. Astringent effect of amaranth also depends on the presence and activity of betacyans. Betacyans belong to the red or purple betalain pigments; the most known is betanidin. These compounds are identified in various species of amaranth [14]. Betalains have recently been recognized as highly bioactive natural compounds with potential human health benefits.

The structure of selected compounds from amaranth oil is presented in Figure 1.

## 4. Supplementary Value of Amaranth

Amaranth seeds have a high nutritional value. The most important product obtained from amaranth is grain, which is a source of flour used in the baking industry [8]. Different plants such as millet, corn, sorghum, pseudocereals (amaranth), quinoa, and teff are the main components of a gluten diet [15]. The lack of gluten fraction makes the amaranth flour suitable for the production of dietetic food (gluten-free products) recommended for people who are allergic to gluten [8,10]. In a recent paper [16], technological and nutritional properties of an innovative gluten-free double-layered flat bread enriched with amaranth flour were examined. New formulations were developed in which rice flour (6%) and starch (6%) were partially replaced with amaranth [16]. Nowadays, such products of good quality are desirable because the number of people with celiac disease is increasing. Amaranth seeds are mainly used to produce flakes, flour, groats and muesli, and oil [7]. The high protein quality of amaranth means that it can be used alone or as a food fortifier in cereal grain mixtures. Recently, amaranth has been also used as a new alternative ingredient to compose functional cookies. The procedure basically relies on partial replacement of whole-wheat flour with formulations based on amaranth flour. The nutritional value of the fortified cookies (with amaranth flour) was found to be higher than that of traditional wheat flour cookies [17]. Oil pressed from amaranth seed is also very popular [12]. Amaranth oil is well known as a functional food [7]. A very important property (advantage) of amaranth oil is that it is highly resistant to oxidation.

During long-term storage, small changes in the fatty acid composition are observed [7]. It is worth noting that the iron content in Amaranth seed is much higher than in wheat, other seeds, spinach, and meat. For this reason, products with amaranth seeds can be an excellent dietary supplement for people with symptoms of anemia [6]. Amaranth and preparations from this plant are eaten in the form of soups, salads, puree, or tortillas [8]. Supplementation with amaranth oil contributes to lowering blood pressure, regulates lipid profile, manifests antioxidant and hepatoprotective effects. Preliminary results of the research indicate that amaranth oil may be used in the normalization of blood glucose levels [7]. Although amaranth is a valuable crop in terms of nutrient content, it is out of reach for many consumers due to its high price. The real problem is also the price of amaranth oil; it is very expensive and therefore long-term supplementation becomes impossible to implement for the average consumer.

Nowadays, with the promotion of healthy lifestyles and taking care of beauty, health-conscious consumers are increasingly choosing a healthy diet with proven health-promoting products and functional foods. An excellent example of such a product is amaranth oil, which, as a rich source of unsaturated fatty acids, tocopherols, and polyphenols, can be an excellent example of a functional food. Moreover, the extracts obtained from amaranth in the vegetation period and early flowering, due to their high content of hydroxycinnamic acid derivatives and rutin, can be a valuable source of antioxidants that can be exploited for the production of nutraceuticals or used as a functional food ingredient [3]. As it turns out, amaranth seeds can also be a source of iron, the amount of which may be important for preventing anemia. The study by Orsango and colleagues clearly presents the conclusion that the consumption of processed bread enriched with amaranth by children in underdeveloped countries decreased anemia prevalence and also increased mean hemoglobin concentration. An in-depth analysis showed that iron deficiency anemia risk was significantly decreased from 35% to 15% in a group of children treated with amaranth [18]. The most important values of amaranth for use as a food supplement are presented in Table 1.

## 5. Status of Amaranth as a Food or Food Ingredient

The seeds, oil, and leaves of this plant are used as food [3,8]. Amaranth seeds were consumed as early as the time of the Incan, Mayan, and Aztec Empires. According to the EU Novel Food Catalogue, in the case of *Amaranthus caudatus, Amaranthus cruentus, Amaranthus hypocondriacus* as food, only the use of grains from the plant is known in the EU. This product was present on the market as a food or food ingredient and was consumed to a significant degree before 15 May 1997, when the first regulation on novel food came into force. Thus, its access to the market is not subject to the Novel Food Regulation (EU) 2015/2283. However, other specific legislation may restrict the placing of this product as a food or food ingredient on the market in some Member States.

## 6. Biological and Pharmacological Activity

This plant has many valuable health benefits. Amaranth has been used as an astringent. This effect probably originates from the content of saponins, protoalkaloids, and betacyans [14]. According to PDR for Herbal Medicines, amaranth has been used for the treatment of diarrhea, ulcers, and in cases of pharyngitis. There are also reports on the use of the plant in excessive menstruation, skin problems such as acne and eczema, and as a mouthwash for sore mouths [19]. Saponins, protoalkaloids, and betacyans are responsible for the pharmacological activity of amaranth [14]. There are reports in the scientific literature regarding the beneficial activity of amaranth on the cardiovascular and nervous systems, hypoglycemic effect, antimicrobial activity, antioxidant activity. Amaranth is widely used in the pharmaceutical industry to produce medicinal products against atherosclerosis, stomach ulcers, tuberculosis, as well as antiseptic, antifungal, and anti-inflammatory preparations [6]. According to Khare 2004, the seeds of *Amaranthus hypochondriacus* L. in Unani medicine are considered as a spermatogenetic drug and tonic. A decoction is used in heavy menstrual bleeding, flowers are treated as remedium for diarrhea, dysentery, cough, and hemorrhages. *Amaranthus polygamus* Willd. is used as a spasmolytic, emmenagogue, galactagogue factor [20]. *Amaranthus spinosus* Linn. is taken to reduce heavy menstrual bleeding and in cases of excessive vaginal discharge, also as a diuretic medium. The whole plants of *Amaranthus blitum* Linn., *Amaranthus gangeticus* Linn., *Amaranthus mangostanus* Linn., and *Amaranthus tricolor* Linn. are considered as astringent, diuretic, demulcent, and cooling [20]. *Amaranthus* tricolor Linn. is placed and described in the Ayurvedic Pharmacopoeia of India. Amaranth seed oil exhibits hypolipemic, anti-atherosclerotic, hypotensive, and antioxidant activity [7]. Therefore, its consumption may lead to inhibition or delay in the development of diet-related diseases of civilization.

### 6.1. Hepatoprotective Activity

Various species of amaranth exhibit hepatoprotective activity. Information on such activity can be found in many scientific papers. Zeashan et al., (2009) demonstrated the hepatoprotective activity of whole plant extract, which was evaluated at 6, 7, 8, 9, and 10 mg/mL concentrations against CCl_4_-induced toxicity in freshly isolated rat hepatocytes and HepG2 cells. Ethanolic extract of *Amaranthus spinosus* showed hepatoprotective activity in a dose-dependent manner [21]. In the study by Aneja et al., (2013), the hepatoprotective activity of aqueous extract of roots of *Amaranthus tricolor* Linn. was analyzed in paracetamol overdose-induced hepatotoxicity in a Wistar albino rat model [22]. The extract examined significantly prevented the physical, biochemical, histological, and functional changes induced by paracetamol in the liver of rats, thereby exhibiting hepatoprotective activity [22]. Other authors also mention the hepatoprotective activity of amaranth, which is attributed to the oil and extracts of the plant [7]. Enrichment of the diet with amaranth oil regulates the lipid profile and has a protective effect on the liver. Primarily, amaranth oil modulates physicochemical properties of lipids and cell membranes of hepatocytes. As a result, it stabilizes cell membranes and acts as a hepatoprotective agent [7]. Squalene is known to exhibit antioxidant and hepatoprotective properties, and also regulates cholesterol levels and helps remove toxic substances from the body [6]. Since a significant squalene content has been found in amaranth oil [13], this liver-protective activity is probably due to this.

### 6.2. Antioxidant Activity

Zeashan and colleagues documented the antioxidant activity of amaranth extract (obtained from the whole plant). In the study conducted by Zeashan et al., (2009), this extract showed significant antioxidant activity in the DPPH assay. In the next study by Lucero-Lopez et al., antioxidant properties of *Amaranthus hypochondriacus* seed extract were also examined. The study was conducted on the liver of rats sub-chronically exposed to ethanol. The results obtained in the experiments confirm the beneficial effect of the tested extract, which as a rich source of polyphenols, had a protective effect on the livers of rats [23]. Sarker and Oba’s work characterized the phytochemical composition of *Amaranthus gangeticus* L. species. They particularly focused on the identification of phenolic compounds responsible for the antioxidant activity of these plants. Twenty-five different phenolic compounds were identified in the plant. Antioxidant components of *A. gangeticus* genotypes exhibited good radical scavenging activities [24]. In another study, the same authors presented chemical compounds found in amaranth *A. tricolor* (betaxanthins, betalains) that exhibit antioxidant activity [25]. In the study by Al-Mamun et al., the antioxidant activity of the methanol extract derived from the seed and stem of *A. hybridus* and *A. lividus* was tested. The DPPH radical scavenging assay showed that both extracts examined possessed significant dose-dependent antioxidant potential, exhibiting IC_50_ values of 28 ± 1.5 and 93 ± 3.23 μg/mL, respectively [26]. In a subsequent scientific paper, two polysaccharides from *A. hybridus* named AHP-H-1 and AHP-H-2 were characterized and examined as potential antioxidant factors. The results obtained in the study confirmed that the two polysaccharides purified from *A. hybridus* have strong antioxidant activity (DPPH radical scavenging activity and superoxide anion free radical scavenging activity) [27]. Kumari and colleagues confirmed the antioxidant properties of another amaranth species, *A. viridis.* Aqueous, chloroform, methanol, and hexane extracts were examined in several in vitro model systems. *A. viridis* exhibited dose-dependent effective antioxidant properties. Major components responsible for his antioxidant activity are gulonic and chlorogenic acids and also kaempferol [28]. In another paper describing the antioxidant activity of amaranth, the phenolic composition of the aerial part of *Amaranthus caudatus* was tested using ABTS+, DPPH, and O_2_ scavenging activity, ferric-reducing antioxidant power (FRAP), and Fe_2_+ chelating ability methods. Different levels of antioxidant activity were observed depending on the stage of plant development and the content of biologically active substances (mainly a wide range of phenolic composition) responsible for generating such activity [3]. Studies focusing on the antioxidant capacity of amaranth over the period 2015–2020 were collected and summarized in Park et al.’s work. In this review, current knowledge on the antioxidant properties of different amaranth species was systematized and consolidated. These properties resulted not only from the presence of phenolic compounds but were also derived from hydrolysates and active peptides with superior antioxidant activity [8].

### 6.3. Anticancer Potential

Water extracts of two amaranth species (*A. lividus* and *A. hybridus*) were examined as anticancer factors. Female Swiss albino mice divided into a few groups were injected with EAC cells and received 25, 50, or 100 μg/mL/day/mouse of test extracts after 24 h of EAC cells injection. The measurement of cancer cells growth inhibition was conducted. Administration of *A. hybridus* and *A. lividus* extracts led to 45 and 43% growth inhibition of EAC cells [26]. The seed extract of *A. hybridus* possessed higher growth inhibitory activity than the stem extract of *A. lividus* and exhibited 14, 26, and 45% growth inhibition at 25, 50, and 100 μg/mL, respectively. In animals treated with amaranth extracts, morphological changes suggestive of apoptosis were also observed in EAC cells. Amaranth preparations can be considered as a potential target for cancer cure studies [26].

### 6.4. Antihyperglicemic and Hypolipidemic Activity

There are scientific papers in the databases on the sugar-lowering and cholesterol-lowering effects of amaranth-containing products. Methanolic extract of *Amarantus viridis* leaves (at the dose of 200 mg/kg and 400 mg/kg per day, 21 days) reduced blood sugar levels in streptozotocin-induced diabetic rats. The administration of the extract also reduced serum cholesterol and triglyceride levels [29]. Girija et al. investigated the anti-diabetic and anti-cholesterolemic activity of the methanol extract of leaves (200 and 400 mg/kg, for 21 days) from three species of amaranth: *A. caudatus, A. spinosus,* and *A. viridis* [30]. Experiments were conducted in streptozotocin-induced diabetic rats. Methanol extracts of all three species of amaranth showed significant glucose and cholesterol-lowering activity at a dose of 400 mg/kg [30]. Similar issues are presented in another paper published in 2011. Antihyperglycemic and hypolipidemic activity of the methanolic extract of leaves of *Amaranthus viridis* was investigated. Normal and streptozotocin-induced diabetic rats were fed with 200 mg/kg and 400 mg/kg of extract *per os* for 21 days. The authors of this study proved that the tested extract showed antiglycemic activity and improved the lipid profile in rats [29]. Studies on the activity of selected proteins from amaranth (*Amaranthus cruentus*) suggest hypocholesterolemic activity of this plant. Manolio Soares and colleagues showed that proteins from the plant affect the action of a key enzyme in cholesterol biosynthesis, 3-hydroxy-3-methyl-glutaryl-CoA reductase [31]. The hypolipemic effect of amaranth oil is associated with its significant squalene content. The mechanism of activity of squalene relies on the inhibition of HMG-CoA activity—a liver enzyme responsible for cholesterogenesis. Such activity has been demonstrated in both rat and clinical studies [7]. In another paper, the effects of consumption of the *Amaranthus mangostanus* on lipid metabolism and gut microbiota in high-fat diet-fed mice were examined. Amaranth powder supplementation significantly reduced the levels of triglycerides, total cholesterol, and phospholipids in the liver of rats and also downregulated the expression of a few lipogenesis-related genes [32]. Recent research findings suggest that the aqueous extract obtained from steamed red amaranth leaves might be used as a potent nutritional supplement to prevent diabetic retinopathy. Anti-glycative and anti-oxidative action of that extract against a high glucose-induced injury was examined in a human lens epithelial cell line HLE-B3 [33].

### 6.5. Neuroprotective and Antidepressant Action

An attempt was made to determine the neuroprotective effect of *A. lividus* L. and *A. tricolor* L. extracts against AGEs-induced cytotoxicity and oxidative stress. Advanced glycation end-products (AGEs) caused oxidative stress and cytotoxicity in neuronal cells. It was found that examined extracts protect human neuroblastoma SH-SY5Y cells against AGEs-induced cytotoxicity [34]. The authors suggest that amaranth may be useful for treating chronic inflammation associated with neurodegenerative disorders [34]. In another paper by the same authors, the neuroprotective effect of amaranth was again described. The methanol extracts of *A. lividus* and *A. tricolor* leaves were found to decrease cell toxicity and intracellular ROS production in human neuronal immortalized SH-SY5Y cells. Examined extracts decreased oxidative stress by suppressing gene expression of HMOX-1, RAGE, and RelA. Because of such activity and the high content of antioxidant substances, amaranth extracts may be a potential neuroprotective factor [35]. The methanol extract of *Amaranthus spinosus* (100 and 200 mg/kg, orally) was investigated for antidepressant activity. In the study, forced swimming test (FST) and tail suspension test (TST) models were used in experimental rats. The results of the tests prove the antidepressive potential of the methanol extract of this plant. The authors indicate that the mechanism of this activity has not yet been understood and its explanation requires further in-depth studies [36].

### 6.6. Anti-Inflammatory Activity

*A. lividus* and *A. tricolor* extracts possess anti-inflammatory activity and can reduce pro-inflammatory cytokine gene expression. An increased amount of proinflammatory cytokines, such as IL-1, IL-6, and TNF was observed [34]. In 2021, information on bioactive peptides with anti-inflammatory activity from germinated amaranth released by in vitro gastrointestinal digestion was described in the scientific literature for the first time [37].

### 6.7. Antimicrobial and Antiviral Effect

A new antimicrobial peptide with strong activity against *E. coli* was found in the medicinal plant *Amaranthus tricolor.* This peptide was selected after analysis of the protein fraction from *A. tricolor* and characterized as being highly antimicrobial [38]. The antimicrobial activity of ethanolic and aqueous extracts of *Amaranthus caudatus* was also examined in a study by Jimoh and colleagues [39]. *Streptococcus pyogenes, Staphylococcus aureus, Bacillus subtilis, Streptococcus pneumoniae, Escherichia coli,* and *Pseudomonas aeruginosa* were tested in this study. The used strains of fungi were: *Candida albicans, Penicillium chrysogenum*, *Candida glabrata*, and *Penicillium aurantiogriseum*. The ethanolic extract of amaranth showed stronger antimicrobial activity than the aqueous extract. The extracts also showed antifungal activity with an MIC in the range of 0.675 to 10 mg/mL [39]. A new application of amaranth seed oil (apolar fraction from *Amaranthus cruentus* L. seeds extract) as an agent against *Candida albicans* was examined by De Vita and colleagues. Amaranth oil in combination with an antifungal drug named terbinafine possesses synergic fungistatic and fungicidal activity and can be a potentially important ingredient of antifungal formulations [40]. In the next study, stem and seed methanol extracts of *A. lividus* and *A. hybridus* were examined as antimicrobial factors. In vitro susceptibility of five pathogenic bacteria (*E. coli, P. aeruginosa, B. subtilis, S. typhi, S. aureus*) was confirmed in the disk diffusion assay [26]. There have also been recent reports of the antiviral activity of amaranth. Chang and colleagues investigated the antiviral properties of betacyanin fractions from leaves of red spinach, *Amaranthus dubius* [41]. Betacyanin fractions from *A. dubius* inhibited DENV-2 in vitro. Betacyanin fractions exhibited antiviral activity against DENV-2 after virus adsorption to the host cells in a dose-dependent manner. For betacyanin fractions from red spinach, the IC50 value was 14.62 μg mL^−1^, with an SI of 28.51. The authors point out that the mechanism of infectivity inhibition by the betacyanins must be confirmed by rigorous scientific studies [41]. In other experimental work, the antimicrobial activity of *A. tricolor* crude extract against *S. aureus* was assessed by disk diffusion, minimum inhibitory concentration (MIC) determinations, and growth curves. The authors of the experiment proved that the extract has antibacterial activity and the mechanism of this activity was connected with cell membrane depolarization, reduction in intracellular pH, decrease in bacterial protein content, DNA cleavage, and leakage of cytoplasm. The plant extract has the potential to be a good food preservative that improves meat quality [42]. The major biological effects of amaranth are summarized in Table 2.

## 7. Amaranth Seed Oil in the Cosmetics Industry

Due to its rich nutritional properties, some amaranth preparations are used in the cosmetics industry. Amaranth oil contains a large amount of unsaturated fatty acids, tocopherols, phytosterols, and squalene. These compounds are beneficial for hair and skin conditions. Amaranth seed oil may be used in the care of all skin types. It perfectly moisturizes, soothes irritations, accelerates wound healing, and has antimicrobial properties. It provides skin-nourishing and anti-aging effects. It contributes to the regeneration, nourishment, and strengthening of the epidermis and acts as an antioxidant. For example, innovative sunscreen formulations based on nanostructured lipid carriers (NLCs) which act as delivery systems for antioxidant and anti-UV bioactives were examined by Lacatusu and colleagues [43]. Amaranth oil and pumpkin seed oil were fitted in the lipid NLCs core, forming new delivery systems that were able to simultaneously entrap UVA and UVB filters and an antioxidant. It is an innovative and non-invasive design of herbal cosmetic formulations with superior photoprotection and enhanced antioxidant properties [43]. Amaranth seed oil is mostly found in skin creams and lotions, and is used as an ingredient in shampoos and shower gels. Amaranth oil as a natural, rich source of tocopherols, protects hair from the harmful effects of sunlight, is an effective way to solve problems associated with greasy hair, strengthens hair, and protects it from excessive hair loss. In addition, oil is also used in beauty clinics. It is usually used in body massages, baths, and relaxation treatments.

## 8. Future Remarks

Amaranth is characterized by many advantages (Figure 2). Therefore, it would be valuable to design a far-sighted cultivation strategy of this plant and to prepare a kind of global campaign promoting the advantages of amaranth, intended for food producers, cosmetics and pharmaceutical companies, and dietary supplement manufacturers. It is also extremely important to promote work leading to the development of new technologies and support research and development activities reflected in the production of food and cosmetics from this plant. This should lead to rapid and effective commercialization of these technologies and their introduction to the market in the form of specific products. In addition, new in-depth scientific research on all types of biological activities of amaranth preparations on human health is still needed.

## 9. Conclusions

The properties of amaranth combine the characteristics of a health-promoting food and a raw material with potential therapeutic activity. Thus, all amaranth products might be utilized as natural agents in the pharmaceutical and food industries. The excellent nutritional value of amaranth and its health-promoting qualities should induce food manufacturers to develop new technologically innovative food products, especially functional foods. In addition, it is necessary to conduct detailed studies on the pharmacological activity of this plant. This will allow to determine the therapeutic doses of this raw material, which can be used in the formulation of medicinal products intended for application in the treatment of specific diseases. Ongoing statistics show that more than 60% of currently used anticancer drugs are related to plant products as their source.

Amaranth is a valuable plant with two faces—it has been a food for centuries, and at the same time in the future, it can be used to produce plant medicines. Amaranth may find wide application in the prevention and treatment of some civilization diseases, such as ischemic heart disease, allergy, type II diabetes, and celiac disease. However, further in-depth activity studies of this plant and preparations obtained from it are required.

Amaranth may also be a key factor in reducing hunger in underdeveloped countries. Amaranth should be recognized as one of the extremely promising nutritional and healthy crops with a great potential to feed the global population. This potential is still under-exploited. Moreover, amaranth preparations are successfully used in the cosmetics industry. This is due to the presence of biological compounds with beneficial nutritional potential in this plant.

## Figures and Tables

**Figure 1 foods-11-00618-f001:**
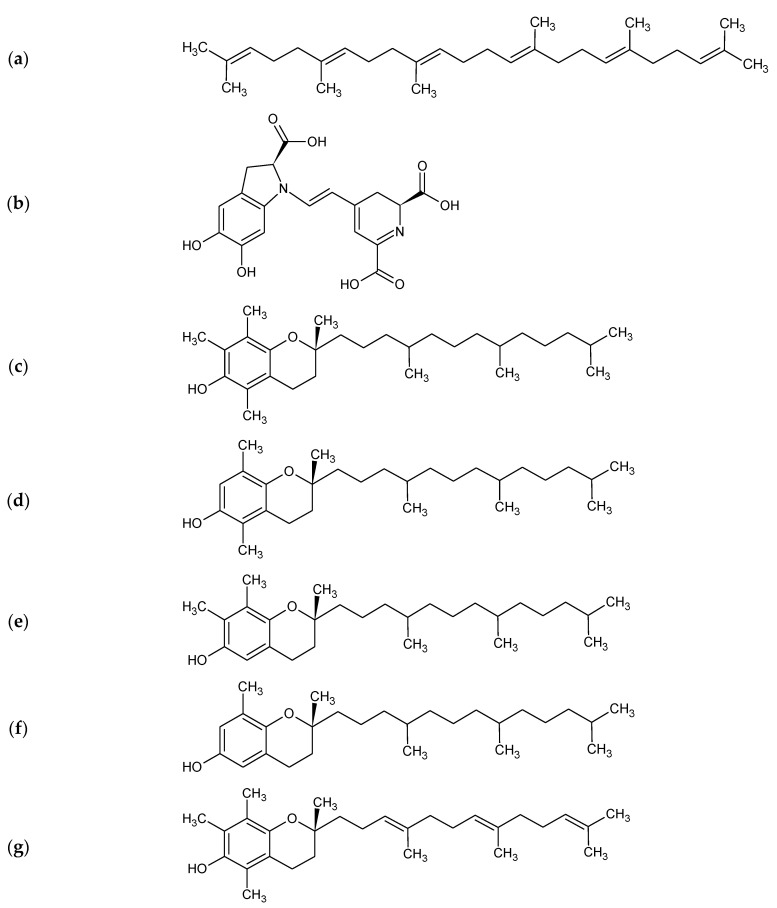
Structure of selected compounds from amaranth oil: (**a**) squalene; (**b**) betanidin; (**c**) α-; (**d**) β-; (**e**) γ-; (**f**) δ-tocopherol; (**g**) α-; (**h**) β-; (**i**) γ-; (**j**) δ-tocotrienol.

**Figure 2 foods-11-00618-f002:**
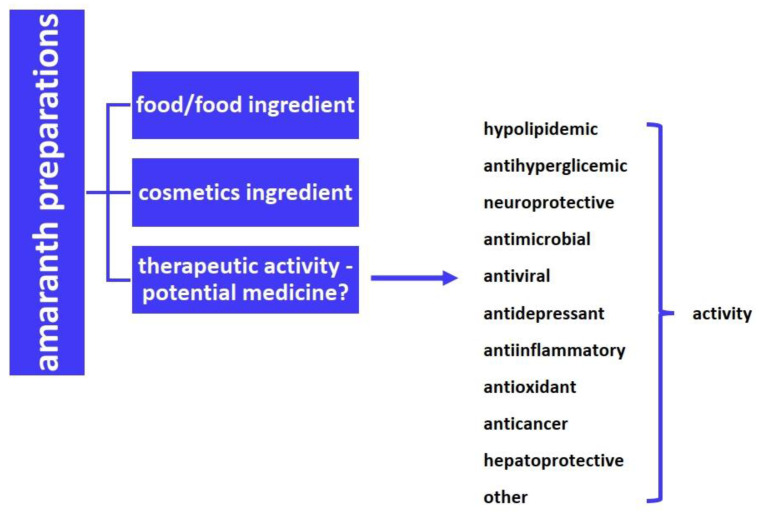
Multiple benefits of amaranth.

**Table 1 foods-11-00618-t001:** Key values of amaranth for use in food supplements.

Content	Source	Main Product Types
Lack of gluten	Seeds, flour	Food, food ingredient
High iron content	Any seed preparations	Food, food supplements
Rich source of unsaturated fatty acids and polyphenolic compound	Seeds, oil	Food, food supplements (nutricosmetics)
Valuable source of antioxidants	Oil	Food supplements (nutricosmetics)
Large amount of squalene	Oil	Food supplements (nutricosmetics)

**Table 2 foods-11-00618-t002:** Key biological effects of amaranth.

Activity	Active Agent	References
Astringent	Saponins, protoalkaloids and betacyans	[14,20]
For skin problems	Naphthalene, squalene, sulfonates of *Amaranthus* spp.	[19]
Hypolipidemic, antihyperglycemic	Methanol extract of *A. viridis* leaves Methanol extract of *Amaranthus* spp. Proteins from *A. cruentus* squalene Leaves aqueous extracts	[29] [30] [31] [7] [33]
Action against microorganisms	*A. tricolor* isolated peptide Ethanolic, aqueous extract of *A. caudatus* Seed oil from *A. cruentus* Methanol extract of *A. lividus* and *A. hybridus* Betacyanins isolated from *A. dubius* *A. tricolor* crude extract	[38] [39] [40] [26] [41] [42]
Neuroprotective or antidepressant	*A. lividus, A. tricolor* extracts *A. lividus* and *A. tricolor* leaves methanol extract *A. spinosus* methanol extract	[34] [35] [36]
Anti-inflammatory	*A. lividus* and *A. tricolor* extracts Bioactive peptides	[34] [37]
Antioxidant	Whole plant extract Seed extract Phenolic compounds Betaxanthins, betalains Seed or stem methanol extract of *A. hybridus* AHP-H-1, AHP-H-2 polysaccharides from *A. hybridus* Different extracts of *A. viridis* Phenolic compounds *A. caudatus*	[21] [23] [24] [25] [26] [27] [28] [3]
Anticancer	Water extract of *A. lividus* Water extract of *A. hybridus*	[26]

## Data Availability

Not applicable.

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
