# Peer review of "The Dual Nature of Amaranth—Functional Food and Potential Medicine"

_foods, 2022, doi:10.3390/foods11040618_

Round 1

Reviewer 1 Report

The paper looks scientifically sound and should addressing the below concerns.

  1. The abstract does not flow well and lacks conclusion and prospect.
  2. Provide a figure containing the structures of the chemical compositions of amaranth.
  3. Provide a table summarizing the key biological effects.
  4. The paper looks like just a collection of data and lacks sufficient critical analysis.
  5. The conclusion section could be supported by elaborated future prospects.

Reviewer 2 Report

In the manuscript entitled, "The dual nature of amaranth - functional food and potential medicine", the authors has showed a brief overview of
current information on the chemical composition of amaranth, the value of its supplementation, the status of amaranth as a food ingredient as well as its main biological and pharmacological activity.

The manuscript is well written and presented. Overall the manuscript should be minor correction in quality of figure-1. 

Reviewer 3 Report

The manuscript addresses the functional properties of amaranth and its potential in medicine.  Amaranth has gained great interest in its consumption during the last years so a document of this type is important, however, a series of suggestions are made to improve its quality.

Line 60-61. Revise the wording “This is a shame” is not a formal expresión and two “because” are in the same sentence. It is recommended to rewrite.

Line 63: “Mamy Countries” correct

Section “Chemical composition of amaranth”. Although the most important nutritional fractions of amaranth are described, it is suggested to add numerical data on their percentages. This is a review and if it is to be multicited it is the type of information a reader would expect to find in the section.

Line 94-99. Add a reference to this information related to the mineral content and other chemicals

Section “Supplementary Value of Amaranth”. It is recommended to add a table with data on the main values of amaranth as a food supplement.

Line 152: Is there any relevance to the date May 15, 1997? Why so much precision in the date? Describe.

Line 159: “Betacyans” these compounds are not described in the composition of amaranth. It is suggested to complement the section on amaranth content.

Figure 1. It is recommended to improve the quality of the figure. As it is the only image in the document, it is suggested to address the subject matter of the document in a broader way. The quality in terms of DPI should also be considered.

Line 183. “Hepatoprotective activity” It would seem that this section needed to be developed. Eliminated if it were the case.

Line 267 y 274 y 279. Check correct name of A. lividus L. and A. tricolor L. and Amaranthus spinosus

Add a section “Future remarks”

When reading the title of the paper, one expects to see two important sections in it. It is suggested to address on the one hand the functional food and on the other hand the "potential medicine". Although these topics are addressed in the document, they are not found in different sections.

Round 2

Reviewer 3 Report

The authors have responded to all recommendations